# "Humane Criminology": An Inclusive Victimology Protecting Animals and People

## Phil Arkow

The National Link Coalition, Etowah, Asheville, NC 28729, USA; arkowpets@snip.net

**Abstract:** To those who primarily associate the word "humane" with "humane society", its connection to criminology might appear to be unrelated. The origins of "humane" and "humane society" are complex and primarily reflect an abiding interest in human and societal welfare rather than animal welfare. Consequently, the origins and evolution of the current American association of humane societies with animal protection—as contrasted to its British association with rescuing victims of drowning—remain shrouded in mystery. A new focus that returns to the original roots of "humane" describing the implications of animal cruelty, abuse, and neglect as cause for human and societal concern due to their potential as sentinel indicators and predictors of interpersonal violence, rather than a strict focus on animals' welfare or their alleged "rights", holds great promise for advancing legislation and community programming that improves the well-being of human and non-human animal species and the prevention of crime.

**Keywords:** humane; humane society; animal welfare; animal rights; child maltreatment; domestic violence; elder abuse





## 1. Introduction

### 1.1. What Does "Humane" Have to Do with Criminology?

To those who primarily associate the word "humane" with "humane society", its connection to criminology might appear to be unrelated. In current American parlance, the terms "humane" in general, and "humane society" specifically, are frequently associated with a primary concern for the welfare of non-human animals. "Criminology" is the human-focused study of crime from a social and academic perspective, examining which individuals commit crimes, why they commit them, their impact, and how to prevent them. Criminology focuses on people, not other animals, explaining criminal behavior as a conscious choice, inherited biological traits, a factor of childhood upbringing and experiences, or community factors such as systemic oppression and informal social controls (Roufa 2020). While corporations can be held civilly and criminally liable for inappropriate actions, the political and economic influence of the agribusiness sector combined with differing perspectives as to whether animals are property or autonomous beings and clashing norms about animal care (Hunter and Brisbin 2016) have often succeeded in getting agricultural operations exempted from many animal cruelty provisions. For the purposes of this discussion, we will focus on individual humans and animals.

In reality, the origins of "humane" and "humane society" are complex and generally reflect an abiding interest in human and societal welfare, including such early social justice issues as penal reform and offender rehabilitation. Even today, the end goal of criminology is said to be to "determine the root causes of criminal behavior and to develop effective and humane means for preventing it" (Roufa 2020). One can readily find examples in mainstream news media of the word "humane" applied to these issues. For example, federal officials in New York, seizing a shipment of 13 tons of weaves made from human hair from people locked up in a Chinese internment camp, said the U.S. was sending a clear and direct message that "illicit and inhumane practices will not be tolerated" (Mendoza 2020).

The Obama administration was embroiled in an issue where an American businessman, a former Iraqi exile named Aiham Al-Sammarae, was sentenced to two years in prison after returning to Iraq; the U.S. Embassy in Baghdad was monitoring the situation to ensure that he was "being treated in a humane manner" (Hosenball and Isikoff 2008). A bill in Virginia to abolish the death penalty called capital punishment "inequitable, ineffective, and inhumane" (Lavoie and Rankin 2021). President Donald Trump's "inhumane" border policy (Ball 2018) of incarcerating immigrant children at America's southern border led a columnist to suggest a solution would be to "modernize or build humane holding facilities" and to spend more money on border security to create "a more humane United States" (Bunch 2018). President Joe Biden sought "a more humane and orderly" system to address the issue (Weissert and Fingerhut 2021).

### 1.2. Origins of a Word

The use of the word "humane" in conjunction with non-human animal well-being is a relatively recent phenomenon. The word is derived from the Latin *humanus* and the French *humaine*, and originated as a common earlier spelling of "human" (Simpson and Weiner 1989); even today, one often finds interchangeable, often incorrect, usages of "inhuman" and "inhumane".

By about 1500 "humane" began to describe gentle, kind, courteous, friendly behavior as befits a human being, with no connotation of protecting animals. In Shakespeare's *Coriolanus*, for example, written in 1608, starving plebeians say that if patricians gave them surplus food "we might guess they relieved us humanely" (I,i); a senator suggests bringing Coriolanus to the marketplace for a public airing of their grievances is "the humane way", whereas executing him would "prove too bloody" (III,i).

After 1700 the word came to describe sympathy with and consideration for the needs and distresses of others (Simpson and Weiner 1989). Samuel Johnson's seminal *Dictionary of the English Language* (Johnson [1755] 1983) makes no association of the word with kindness to animals, rather defining humane as "kind, civil, benevolent, and good-natured".

Throughout the 17th through the 19th Century, one finds continual use of the word "humane" to reflect these qualities. For example, the first attempt to start a federated fundraising drive in the U.S. was undertaken in Philadelphia in 1829 by one Matthew Carey, who entreated 97 "citizens of the first respectability" to sign an appeal entitled, "Address to the Liberal and Humane" (Cutlip 1990). In an 1829 letter by President Andrew Jackson directing the forced relocation of five American Indian tribes from Mississippi and Alabama in the infamous "Trail of Tears", Jackson told Indian leaders that if they wanted to preserve their nation the relocation plan was the only way by which "they can expect to preserve their own laws, and be benefitted [sic] by the care and humane attention of the United States" (Colimore 2009). An 1840 morality tale describes how a couple in 1745 "humanely" took it upon themselves to care for the three children of neighbors who were imprisoned and executed during the Scottish insurrections (Mrs. Blackford 1840).

Other early American usages of "humane" were similarly humanitarian and humanocentric in scope but with no reference to kindness toward animals. The word was found in early calls for penal reform, such as advanced by William Penn who in 1681 created "a more humane house of correction based on labor," and by Benjamin Franklin who in 1790 advocated for "humane treatment of inmates" in Philadelphia's Walnut Street Jail by instituting solitary cells (Dickinson 2008).

One of the more curious usages was the creation of "Humane Fire Companies" that were established in Philadelphia, Easton and Norristown, Pa., and Bordentown, N.J., as early as 1797. The Pottsville, Pa. Humane Fire Co. No. 1, founded in 1829, is still in service today in a fire station located on Humane Avenue (Arkow 2010a).

### 1.3. A Relatively New Phenomenon

The addition of animals and animal welfare into the "humane" movement is a relatively recent phenomenon primarily limited geographically to the United States and

Canada. Building on philosophical concepts expressed by Pythagoras, Porphyry, Thomas Aquinas, Montaigne, and John Locke, among others, that there is a nexus between cruelty to animals and future interpersonal violence, 19th Century moralists and advocates for social reform framed animal abuse within the context of related social reform movements. In the years after the Civil War, independent societies for the prevention of cruelty to animals (SPCA) emerged following the success of antebellum social causes such as the abolition of child labor and slavery, the temperance movement, the elimination of debtors' prisons, and prison and asylum reform (Unti 2008).

The fledgling SPCA movement took off in three parallel strategic directions as evidenced in the nation's most prominent cities. In New York City, the American SPCA (ASPCA), founded in 1866, focused on law enforcement and prosecution of offenders; in Boston, the Massachusetts SPCA, founded in 1868, emphasized educational outreach to prevent animal abuse; and in Philadelphia, the Women's Pennsylvania SPCA, founded in 1869 when it split off from the Pennsylvania SPCA that would not allow women to serve on its board of directors, pioneered the concept of sheltering animals.

For reasons that remain unclear, many of these early societies for the prevention of cruelty to animals soon changed their names to "humane societies". It is possible that this name change reflected a broader approach to the prevention of violence that included the prevention of, and response to, child abuse and neglect. A popular myth arose following the widely publicized "Little Mary Ellen" case in 1874 of an indentured child in New York City rescued by the ASPCA. Although there had earlier been limited cases of criminal charges brought against parents abusing their children, they were largely unenforced (Watkins 1990; Pecora et al. 2000). This incident, in which the girl was placed into protective custody and which led to the founding of the first society for the prevention of cruelty to children, coincided with an encompassing concern for equity and social justice and an attempt to use new Darwinian thought to lessen the distance between humans and animals (Costin 1991). However, the formation of the Oregon Humane Society (1868) and the Humane Society of Missouri (1870), prior to the "Little Mary Ellen" case (Fox 1989), through their organizational name changes appear to belie this theory. Nevertheless, many SPCAs not only changed their names but also their purviews and incorporated child protection into their activities.

Informally organized in 1868, a newspaper account in 1873 announced the formation of the Oregon Society for the Prevention of Cruelty to Animals, by "humanitarians" and "humanely disposed persons", actuated by "noble and Christian motives", to prevent cruelties to animals and to stimulate and generally encourage "a humane and considerate treatment of the lower orders of animal beings". The article added that "every humane citizen will rejoice that the ball has at length been set in motion" in reflecting "the humane and Christian sentiment" of the community (Oregon Humane Society, personal communication, 12 May 2016). The Oregon Secretary of State's Articles of Incorporation on 30 August 1880 certify the incorporation of the organization, however, as the Oregon Humane Society, the name by which it is still known today.

Beginning in the 1870s, child welfare and animal welfare work often overlapped: pioneering and muckraking social reformer Jacob Riis (1892) described the American Humane Association, organized in 1877 as a federation of local organizations, as protecting "the odd link that bound the dumb brute with the helpless child in a common bond of humane sympathy" (p. 150). Child protection work closely aligned the new animal protection movement with other social reform and social justice movements concerned with cruelty, violence, and the social order (Unti and DeRosa 2003; Arkow 1992). For example, the Illinois SPCA, founded in 1869, changed its name in 1877 to the Illinois Humane Society to more accurately represent an expanded focus that had come to include the prevention of cruelty to children (Hubbard 1916). By 1922, 307 of the American Humane Association's 539 animal protection organizational members devoted their work to the protection of abused children as part of the same humanitarian continuum of care (Shultz 1924). Clifton (2014) observed that for many years until the Great Depression, more humane societies operated

orphanages than ran animal shelters, usually as a secular alternative to the work long done by religious organizations and the infamous public workhouses described by Dickens. The Connecticut Humane Society, the last humane society to handle child protective services in lieu of a government agency, did not relinquish that role until 1966.

*1.4. Origins of "Humane Society"*

Persons outside the animal welfare field are frequently confounded by the number and variety of private, nonprofit animal welfare organizations operating in their community and on the national level. Many of these organizations are called humane societies, societies for the prevention of cruelty to animals (SPCAs), animal rescue leagues, animal sanctuaries, and related terms; while their operations differ from organization to organization, their missions are generally similar and the terms are frequently used synonymously. Some SPCAs have added "humane society" to their legal names, and vice versa, in part to capture potential bequests made non-specifically to "the humane society" or to "the SPCA". Some organizations have chosen their names so as to not be confused with existing organizations, or to emphasize a primary function such as sheltering, adoption, rescue, or cruelty prevention. To the non-professional, the nomenclature is often hopelessly bewildering. Further confusing the public, local humane societies are not a branch of the Humane Society of the U.S., nor are local SPCAs affiliates of the ASPCA.

Curiously absent, however, from several early histories of the animal welfare humane movement (Unti and DeRosa 2003; Fox 1989; Niven 1967; Swallow 1963; Fairholme and Pain 1924; Shultz 1924; Stillman 1913), is an explanation as to how the generic term "humane society" came to be associated with animal protection. While it would seem self-evident to North Americans that a "humane society" is synonymous with animal protection, a look at the United Kingdom—where the SPCA concept was invented in 1824—reveals a different picture, where the Royal Humane Society and its offshoots have been rescuing human drowning victims since 1774, and where non-human animal welfare work is done under the aegis of the Royal SPCA. Even today, a "humane society" in the U.K. and Australasia refers to an organization that rescues drowning victims, teaches water safety, and recognizes individuals for acts of bravery in saving victims from drowning or fire.

This interpretation of "humane society" appears to have begun with the formation in 1708 of China's Chinkiang Association for the Saving of Life that established lifesaving stations and lifeboats on the Yangtze and Min Rivers as both government-funded and privately-sponsored services. This concept of human lifesaving organizations came to Europe with the formation in 1767 in Amsterdam of the Society for the Recovery of the Drowned to cope with the common problem of people perishing in the city's canals (Moniz 2008). Fears of premature burial, fueled by actual cases and folk tales, were widespread across Europe in this era, and were subsequently aggravated by even greater fears of "a fate worse than death"—that individuals presumed to be dead, but who were still merely unconscious, might be treated as corpses who could be stolen and used as anatomical objects for dissection in the new science of medical education (Richardson 1988).

By 1774, the Dutch organization had been replicated in England with the formation of the Royal Humane Society in London. This organization was founded in 1774 by two doctors, William Hawes (1736–1808) and Thomas Cogan (1736–1818), who were concerned about the number of people wrongly taken for dead—and, in some cases, buried alive. Both men wanted to promote the new, but controversial, medical technique of resuscitation and offered money to anyone rescuing someone from the brink of death (Royal Humane Society 2021). The British organization claimed in its first decade to have "restored to their friends and country" 790 out of 1300 persons apparently dead from drowning (Humane Society of the Commonwealth of Massachusetts 1786). Similar lifesaving "humane societies" were soon founded in the British Isles, such as the Humane Society of St. John in Ireland. This organization in 1819 offered to fund a watch for the Hospital Fields burial ground in Dublin, which held the largest concentration of paupers' graves

in the city and was known as a favorite target for body snatchers. The Humane Society's charitable gesture was sharply rebuked by a professor of anatomy at Trinity College who protested that the watch would disrupt the medical school, which brought £70,000 worth of business a year to Dublin and whose cadavers were used not only for experimentation but also provided false teeth for the upper and middle classes (Richardson 1988).

With the expansion of the British Empire, the Royal Humane Society concept spread to member Commonwealth nations. The Victorian Humane Society was inaugurated in Melbourne, Australia in 1874 to honor the founding group's centennial: it was renamed the Royal Humane Society of Australasia in 1882 (Royal Humane Society of Australasia 2021) and was the impetus for the formation in 1898 of the Royal Humane Society of New Zealand to bestow awards upon those who risk their own lives to save the lives of others in peril (Royal Humane Society of New Zealand 2021). Similar bravery-recognition groups exist locally, such as the Royal Humane Society of New South Wales in Australia and the Liverpool Shipwreck and Humane Society in the U.K.

The Royal Canadian Humane Association, called "Canada's Bravery Awards Association", was founded in 1894 "to recognize such deeds of heroism, by Canadians in civilian life, who, through their alertness, skill and concern, save or attempt to save a life, especially where those actions lie outside the ordinary duties of the person involved" (Royal Canadian Humane Association 2021), although there are animal welfare "humane societies" as well today in Canada, such as the Ottawa, Toronto and Calgary humane societies, among others. "Humane society" is still quite limited when referring to an animal shelter in Australia and New Zealand, although animal welfare advocates in those nations are familiar with the term as an import from the U.S.

Following the Dutch and English experiences, similar water-rescue and resuscitation groups quickly were initiated in Paris, Venice, Hamburg, Milan, and eventually came to the U.S. In 1785, a group of Boston citizens, concerned about needless deaths resulting from shipwrecks and drowning, met several times at the Bunch of Grapes tavern to consider starting an organization. Formally established in 1786 and incorporated in 1791, The Humane Society of the Commonwealth of Massachusetts elected James Bowdoin, the governor of Massachusetts and the founder of Bowdoin College, to be its first president (Humane Society of the Commonwealth of Massachusetts 2021).

This organization was founded "for the recovery of persons who meet with such accidents as to produce in them the appearance of death, and for promoting the cause of humanity, by pursuing such means, from time to time, as shall have for their object the preservation of human life and the alleviation of its miseries" (Humane Society of Massachusetts 1845). John Lathrop (1787) praised the nascent Humane Society of Massachusetts as being the first benevolent institution to address "the cries of the needy", "the fight of wretchedness", and "the relief to prevent misery" among those suffering from apparent death. It exists today as the third oldest charitable society in that state.

Similar human lifesaving humane societies soon emerged in Philadelphia, New York, and elsewhere. Many of these groups erected rescue sheds and boathouses along the Atlantic seaboard and published guidelines for the "reanimation" and resuscitation of persons who appeared to have died from drowning, heat prostration, hypothermia, lightning strikes, and other causes (Humane Society of Philadelphia 1788). These organizations became the forerunner of the U.S. Life-Saving Service which evolved into the Revenue Marine Corps, later the Revenue Cutter Service, and today is known as the U.S. Coast Guard (Shanks et al. 1996).

When the work of the Humane Society of Philadelphia, founded in 1780 to reanimate and resuscitate drowning victims, began to diminish, it was disbanded. Its mission, however, was taken over in 1861 by the Philadelphia Skating Club and Humane Society to rescue ice skaters who fell through the ice on the city's frozen rivers (Arkow 2017).

Meanwhile, other early American humane societies began to promote the aforementioned "cause of humanity" and "the alleviation of its miseries". They fought for prison reform and temperance. They tried to close "petty taverns and grog-shops"

which, while seen as being necessary for the weary traveler, were called "the nurseries of intemperance, disorder and profligacy" among the laboring poor of New York City (Humane Society of the City of New York 1810).

## 2. The Animal Connections: Three Competing Philosophies

### 2.1. Humane Societies and Animal Welfare

Somehow despite, or perhaps as a result of, these humanitarian causes, "humane society" came to be associated with animal welfare in North America. An unnamed "humane society" was cited by Thoreau in *Walden* (Thoreau [1854] 1971) as being the greatest friend of hunted animals (p. 211). "No humane being, past the thoughtless age of boyhood, will wantonly murder any creature which holds its life by the same tenure that he does", Thoreau wrote (p. 212), although elsewhere he apotheosized Nature when the winds sigh "humanely" (p. 138) and described philanthropy as a "humane" pursuit (p. 73). In *Cape Cod* (Thoreau [1865] 1993), Thoreau described the "Charity or Humane Houses" erected on the beaches of Barnstable County where shipwrecked seamen may look for shelter, apparently referencing the Humane Society of Massachusetts' water-rescue boathouses. But in the same book, in mentioning an incident where a little boy had poached 80 swallows' eggs from their nests, Thoreau wrote, "Tell it not to the Humane Society". Although the Humane Society of the Commonwealth of Massachusetts later expanded its sphere of interest to help establish the Asylum for the Insane in 1816, to support the Boston Lying-In Hospital and to reward individuals for heroic personal rescues with medals and monetary awards, there is no apparent indication that the protection of wild birds was a matter of organizational concern (Massachusetts Historical Society 2021).

Whatever and whenever the historical origins of humane societies beginning to address animal concerns, the humane movement today has expanded into three parallel, and sometimes conflicting, philosophies in contemporary animal protection, amid several other views of human-animal relationships (Morgan 1983) which often serve to confuse the general public. The traditional roots of the humane movement have been to promote *animal welfare*—a philosophical construct that humans have a moral obligation to care for other animals. The historical antecedents of this philosophy are rooted in long-standing human-centered concerns either about the loss of use of animals as property, or consideration of what additional harm the perpetrators of such acts might do to humans (Arkow and Lockwood 2016).

Early writings on duties to animals did not consider the impact of cruelty toward animal victims, but rather the potential for acts of animal abuse to be precursors to interpersonal violence or antisocial behaviors. In *Summa Contra Gentiles,* St. Thomas Aquinas reflected this prevailing view:

> . . . if any passages of Holy Writ seem to forbid us to be cruel to dumb animals, for instance to kill a bird with its young, this is . . . to remove man's thoughts from being cruel to other men, and lest through being cruel to other animals one becomes cruel to human beings . . . (St. Thomas Aquinas, as cited in Regan and Singer 1976, p. 59)

Some five centuries later, Immanuel Kant echoed this concern about animal cruelty's adverse impact on the human condition in his essay *Metaphysical Principles of the Doctrine of Virtue:*

> . . . Our duties towards animals are merely indirect duties towards humanity. Animal nature has analogies to human nature, and by doing our duties to animals in respect of manifestations of human nature, we indirectly do our duties to humanity. . . . cruelty to animals is contrary to man's duty to himself, because it deadens in him the feeling of sympathy for their suffering, and thus a natural tendency that is very useful to morality in relation to other human beings is weakened. (Kant, as cited in Regan and Singer 1976, p. 125)

[Davis](2015) observed that American animal protectionists from earlier centuries who strove to prevent pain and suffering through their commitment to animal welfare might seem unrecognizable today. Most of them ate meat, believed in euthanasia as a humane end to suffering, and justified humanity's kinship with other animals through Biblical ideas of gentle stewardship. They accepted animal labor as a compulsory burden of human need before the advent of the Industrial Revolution and motor vehicles.

The Biblical antecedent (e.g., the guidance in Genesis 1:26 establishing that humans have dominion over animals but that this dominion is tempered with a moral responsibility to care for them) was manifested in linkages between the early humane movement and Protestant revivalism and social reform in the early 19th Century. These, in turn, spearheaded the development of early anti-cruelty laws. The United States is believed to have the world's oldest laws specifically for the prevention of animal cruelty, legislation that even predates the founding of our nation. Animal protection entered the American colonial record in December 1641, when the Massachusetts General Court enacted its comprehensive legal code, the "Body of Liberties". Sections 92–93 prohibited "any Tirranny or Crueltie towards any bruite Creature which are usuallie kept for man's use" and mandated periodic rest and refreshment for any "Cattel" being driven or led. Puritan animal advocates believed that cruel dominion was a consequence of Adam and Eve's fall from the Garden of Eden; kindly stewardship, however, reflected their reformist ideals, thus illuminating a long historical relationship between religion, reform, and animal protection ([Davis 2015](#)).

Throughout the Second Great Awakening (1790–1840), Protestant ministers and evangelicals embraced a new theology of free moral agency and human perfectibility which included mercy toward animals. Animal welfare was linked with abolitionism and the temperance movement as a barometer for human morality and kindness to animals—especially through a "gospel of kindness" as taught to children—as a moral complement to sobriety and the marker of a more advanced civilization ([Davis 2015](#)). Many humane societies and SPCAs, following the Massachusetts SPCA model, embraced the education of children as the most effective strategy for achieving kindness to animals.

George Angell, founder of the Massachusetts SPCA and later its offshoot American Humane Education Society, stressed humane education's utility for ensuring public order, suppressing anarchy and radicalism, smoothing relations between the classes, and reducing crime; it would be a valuable means for socializing the young (especially of the lower socioeconomic classes) and the solution to social unrest and revolutionary politics. The promotion of humane education as an antidote for depraved character and a panacea for societal ills aligned the fledgling animal protection movement with other social reform and justice movements concerned with cruelty, violence, and the social order ([Unti and DeRosa 2003](#)). The Latham Foundation, founded in 1918 for the promotion of humane education, exemplifies this paradigm. A poster from the 1930s, still in use today, depicts two children with a puppy approaching a set of steps leading to "world friendship". The first step up this hill is "kindness to animals", which will subsequently take the voyagers to kindness to each other, other people, our country, other nations, and the world ([Arkow 2019](#)).

This concept was widely popular in the Victorian Era, particularly once the evangelical and social justice movements' social reforms benefiting the vulnerable and victimized, such as slavery, women's rights, prison reform, conditions for factory workers, and the care of the insane, were under way. Angell's "humane education" was seen as a means of insulating youth, and boys in particular, against tyrannical tendencies that might undermine civic life were their violent natures to go unchecked. The suffering animal was seen as a particularly noble and selfless servant, nicely suited for instruction and an important means of inculcating such standards of gentility as self-discipline, Christian sentiment, empathy, and moral sensitivity. Humane education was infused with societal class stratification as well, as its advocates saw the teaching of "kindness to animals" as a way to separate refined, urbane, middle- and upper-class standards from the coarser, rustic behaviors of lower

classes and immigrants who were considered the sources of much brutality (Angell 1892; Unti and DeRosa 2003; Ritvo 1987; Grier 2006; Turner 1980).

One finds these sentiments about animal cruelty being a precursor to antisocial behaviors throughout Victorian Era children's literature. For example:

> "A worm, a fly, and all things that have life, can feel pain: if we learn to be cruel while boys, we shall not grow up to be good men" (Cobb 1832)

> "One who is cruel to a cat or a dog, a bird or a fish, will be cruel to his fellow-man, and such cruelty dulls all those finer feelings which make a true gentleman or lady" (Johnson 1900)

### 2.2. Animal Control

The ASPCA's state charter in New York in 1866 gave the fledgling organization police powers to prosecute animal abuse, particularly the welfare of horses in urban areas. Although anti-cruelty laws had been enacted in other states as early as the 1820s, the ASPCA model of private, nonprofit animal advocates empowered to enforce laws soon became a viable system in other parts of the country (Arkow 1991).

By the latter years of the 20th Century, this approach would fuel the development of a second philosophy of animal protectionism, *animal control.* In many communities, stray animals had been rounded up by "dogcatchers", law enforcement or municipal officers empowered to protect the public from stray dogs amid growing awareness of the threat of rabies; Wild Bill Hickok was paid 50 cents for every stray dog he shot on the streets of Abilene (Arkow 2010b). In an era before rabies vaccines and population-controlling pet sterilization became available, local dogcatchers staged massive summertime roundups in which strays were shot or violently thrown into crowded wagons and killed at the pound (Davis 2015). Arkow (1987) described the difference between animal welfare and animal control as two sides of the same coin: animal welfare protects animals from people, and animal control protects people from animals.

Despite a history of "humane oriented dog sheltering" affecting kennel welfare and dog management dating back to at least the 14th Century, and community pounds in medieval Europe and the American colonies that provided humane care to stray livestock (Smith-Blackmore 2017), urban American dog pounds were often described as horrid, with methods of euthanasia that were frequently far from humane. In a particularly ironic example, although humane societies were founded to rescue people from drowning, many communities employed drowning as a way to rid cities of unwanted dogs. In New York City, as many as 48 unwanted dogs were lowered at any one time in barred crates into a river and submerged for 10 min (Smith-Blackmore 2017).

Having animal protection advocates provide animal control and/or sheltering services, usually under contract to municipal or county governments, was often seen as a more humane alternative than dispassionate municipal employees whose motives may have been more directed by a bounty system than an overarching concern for humane treatment of animals (Arkow 1991). In Philadelphia, the Women's Branch of the Pennsylvania SPCA instituted new capture methods and transformed Philadelphia's municipal pound into a humane "shelter", where dogs received regular care. The ASPCA was finally given control over New York's stray animal population in 1894 and maintained a contract for those services for more than 90 years. Many mainstream SPCAs and humane societies across the U.S. eventually incorporated stray management, sheltering, and adoption into their programming.

By the end of the 20th Century, however, this contract-for-services model began to fall out of favor. Municipalities' growing concern over the liability of outsourcing law enforcement powers to advocates who were not sworn peace officers, and humane organizations recognizing that their service contracts often failed to adequately cover the costs of providing those services, caused many humane groups to divest themselves of this function and revert back to a purely animal welfare model, focusing instead on animal adoptions, veterinary care, community education, and advocacy. Many municipal and

county animal control operations became strictly governmental functions, albeit still an underfunded stepchild in relation to other municipal departments who often regard animal issues as less significant than human welfare concerns.

Through training mechanisms advanced by groups such as the National Animal Care and Control Association, other national organizations, and state animal control associations, animal control came to be seen as more professional. Like the early humane movement, today's animal control officers see themselves with a dual focus of protecting animals and people. However, even today many animal control officers are not sworn peace officers; lacking powers to enforce misdemeanor or felony animal cruelty issues, they are forced to limit the scope of their work to stray animal roundup and impoundment, and license and rabies vaccination verification. There is no consistent organizational structure: they may be employed by law enforcement, code enforcement, health department, a separate free-standing agency, or even the occasional fire department (National Link Coalition 2021a). In some communities their purview is further limited to only to dogs and they may not interact with cats or wildlife (Smith-Blackmore 2017). This fragmentation of services and response and the absence of either a single, uniform method to report suspected crimes against animals or a national or state coordinating agency, as is common in child protective services, cause tremendous confusion among the general public. The National Link Coalition (2021a) compiled a directory of responding agencies in over 6500 cities and counties; whereas each state may have only one or two hotlines to report suspected domestic, child or elder abuse, some states list more than 600 different entities to investigate suspected animal cruelty, abuse, or neglect.

### 2.3. Animal Rights

Meanwhile, the coalition of movements dedicated to moral uplift that had given animal protection its interconnected human and animal welfare agenda eventually fractured, leading to one segment with a singular focus on animals. The professionalization of social work cleaved the earlier union of child and animal protectionists into separate fields, and the passage of federal legislation created a government-mandated national system of child protective services (Davis 2015). One offshoot of this has been the irony that despite the emergence of child protection as a function of animal protection work, only 27 out of 889 accredited BSW and MSW programs in the U.S. have been identified that reference the human-animal bond in pre-professional training curricula (National Link Coalition 2021b). The repeal of the 18th Amendment in 1933 dissolved the temperance movement and gradual secularization further transformed animal protection (Davis 2015).

Energized by the social justice movements of the 1960s and 1970s, the traditional *animal welfare* paradigm began to be superseded in the latter third of the 20th Century by a new focus on *animal rights*—a philosophy that non-human animals have inalienable rights in and of themselves.

This concept of animals having inherent legal rights—as opposed to animal welfare's emphasis on a moral obligation by humans to care for animals who are property and consequently lack legal standing—had been espoused for some time. The idea that animal cruelty is an intrinsic, moral evil based on its impact on the animal itself first gained a foothold in the late 18th century. This view was expressed by Rev. Humphry Primatt in *A Dissertation on The Duty of Mercy and Sin of Cruelty to Brute Animals* (Primatt [1776] 2010):

> "Pain is pain, whether it be inflicted on man or on beast; and the creature that suffers it, whether man or beast, being sensible of the misery of it while it lasts, suffers evil" (p. 21)

One of the most influential ideas concerning animal's alleged rights was buried in a footnote to *Introduction to the Principals of Morals and Legislation* by Jeremy Bentham (1789) entitled "Interests of the inferior animals improperly neglected in legislation". Bentham argued that the capacity for suffering is the vital characteristic that gives a being the right to legal consideration. The final sentence of the footnote is often used today as a foundation

for those seeking to promote the cause of animal rights: "The question is not, Can they reason? nor, Can they talk? but Can they suffer?" (p. 130).

First popularized by Singer (1975), who argued that sentient creatures have a right to "equal consideration" because they can suffer, and who considered "speciesism" to be a form of discrimination akin to racism and sexism, this notion was further expanded by Regan (1983) who contended that animals possess intrinsic moral rights as individual "subjects of a life" with complex feelings and experiences that extend beyond their ability to suffer.

The result of these and similar publications was to create a new animal rights paradigm that diverted the humane movement away from its original species-spanning work of protecting children and animals and caring for social reform into a singular focus on only non-human animals. This, in turn, helped inspire a movement towards "no-kill" animal shelters operated by individuals struck by the irony that humane organizations originally created to save horses were now operating animal shelters that were euthanizing inordinate numbers of unwanted and stray dogs and cats (Caras 1983).

The animal rights focus, while highly popular, encountered a formidable if not impenetrable obstacle in legal theory and history: animals are chattel property and arguably will continue to be legally defined as such for the foreseeable future. Although cruelty obviously has adverse effects on the animals themselves, animal cruelty is treated as a crime against the property owner or against society. The status of animals as a special category of property that is capable of pain, suffering, and death, and to which we can form unique attachments, is rarely recognized in legislation or case law. However, this special status is not totally ignored. It is often what leads the courts to take animal cruelty more seriously in consideration of what the act tells us about the perpetrator and the risks he or she poses to society (Arkow and Lockwood 2016).

Furthermore, passionate animal rights advocates' efforts to enact favorable legislation run into twin walls of reality: the vested financial and commercial interests and political lobbying power of agribusiness which far exceed that of a fragmented, highly emotional community not familiar with the legal and criminal justice systems; and the priorities of budget- and time-strapped federal, state, and local legislators for whom the moral concerns of animal well-being are not as significant as the protection of human safety and health. For example, a law enacted in Florida in 2021 that requires veterinarians to report suspected animal abuse to law enforcement authorities specifically exempts such reporting if the alleged incident occurs at a commercial food-producing animal operation (National Link Coalition 2021c).

## 3. "Inclusive Victimology": Linking Animal Abuse and Human Welfare

Confronted with both this tri-partite fragmentation of the animal protection community and a silo effect where animal care and control agencies have been marginalized rather than being considered equal partners with community human social services agencies, a new direction began to emerge in the 1990s. This new direction links these disparate elements by returning to the original roots of the movement. Rather than viewing animal's well-being as either a matter of moral concern, a result of their alleged inalienable rights, or an incidental activity of municipal code enforcement, what has come to be called "the Link" sees acts of animal abuse as "red flag" potential precursors or predictors of incipient or concurrent antisocial behaviors and interpersonal violence, offering opportunities for earlier intervention. The Link views a coordinated, multi-disciplinary response as the most effective strategy to pool limited organizational resources, avoid unnecessary duplication of services, and prevent four forms of family violence: child maltreatment, domestic violence, elder abuse, and animal cruelty, abuse and neglect, as all four may involve common motivations, risk factors, perpetrators, and victims (Arkow 2019).

This approach came into focus in 2008 with the formation of the National Link Coalition (2008), an ad hoc, multidisciplinary network of representatives from animal care and control, veterinary and human medicine, prosecution, law enforcement, child and

adult protection, domestic violence, and academia. Motivated to create safer communities through collaborative inter-agency cross-training and cross-reporting efforts to protect all vulnerable members of the family, the Link uses extensive empirical academic research and anecdotal experiences to expand a basic premise that "when animals are abuse, people are at risk, and when people are abused, animals are at risk" (Arkow 2019).

A Link approach demonstrates that "humane" and "criminology" are not mutually exclusive, as the former word has been associated with human behaviors far longer than it has been with animal protection. A new look at "humane criminology" should focus on the commonalities between animal abuse and other crimes to find root causes of species-spanning criminal behaviors and to develop effective means for preventing them.

Keeley (2020) suggested that a more comprehensive and contemporary examination of humane behavior, and its impact on criminology and the law enforcement and criminal justice systems, would be to follow up Bentham's statement of "The question is not, Can they reason? nor, Can they talk? but Can they suffer?" with another question: "Who else could be suffering by the same hands?" She added, "Too often we focus on the species of the victim when instead we should focus on the abusive nature of the perpetrator. The voiceless and the vulnerable deserve our attention, so that no victim is left behind" (p. 25).

Turgoose and McKie (2020) called for "inclusive victimology" in which the criminal justice system replaces an anthropocentric look at humans as victims and animals as property with a more ecological perspective that recognizes the interconnectedness of people, animals, and the environment. In describing how the coercive control tactics employed by domestic violence abusers to intimidate and retaliate against their partners often victimizes pets in "emotional blackmail", they noted how both humans and pets can experience physical and emotional victimization in domestic abuse scenarios.

The Link focus has attracted considerable interest in the law enforcement and criminal justice sectors. Guidebooks and articles to help professionals understand how violence against animals can predict or be associated with concurrent interpersonal violence have been published by numerous organizations, including the National District Attorneys Association (Phillips 2014; Phillips and Lockwood 2013); the American Prosecutors Research Institute (Lockwood 2006); the International Association of Chiefs of Police (Lockwood 1989, 2000; Palais 2020; Ponder and Lockwood 2001; Turner 1980); the U.S. Department of Justice's Office of Community Oriented Policing Services (Lockwood 2012; National Sheriffs Association 2018); the federal Joint Counterterrorism Assessment Team (2018); the U.S. Office of Juvenile Justice and Delinquency Prevention (Ascione 2001); the National Council of Juvenile and Family Court Judges (Balkin et al. 2019; NCJFCJ National Council of Juvenile and Family Court Judges); the American Bar Association (Davidson 1998); the National Sheriffs Association (Doherty and Smith-Blackmore 2017; Layton 2019; Randour 2013; Thompson 2019); and innumerable law journals and state associations representing these professions. The Hon. H. Lee Chitwood, of the Juvenile and Domestic Relations District Court in Pulaski, Va., described one judge's reaction to the Link:

> "Animal cruelty does not occur in a vacuum, and the failure to fully examine its origins would likely lead to future criminal acts and the continued cycle of abuse and violence" (Animal Legal Defense Fund 2021b)

## 4. Data: Cruelty to Animals as a Precursor or Co-Occurring Factor with Other Crimes

The National Link Coalition has compiled a bibliography (Animaltherapy.net 2021) of over 1500 references describing the potential for animal cruelty, abuse, and neglect to be a sentinel precursor to, or co-occurring incident with, other forms of family and community violence. A sampling of some of the more salient findings include:

### 4.1. Animal Abuse and Domestic Violence

- A crisis line that identified harm or threats to animals, access to weapons, and suicide threats as key risk factors for domestic violence homicide saw the number of femicides decrease 80% (Boat and Knight 2000).

- 41% of intimate partner violence offenders had histories of animal cruelty (Febres et al. 2014).
- Domestic violence batterers specifically choose pets as soft targets because they believe the police do not care about animal cruelty and they can get away with it (Roguski 2012).
- 76% of domestic violence victims whose partners had histories of pet abuse had been strangled, 26% had been forced to have sex with the suspect, and 80% feared that they would be killed by the suspect. When perpetrators of intimate partner violence also have a history of animal abuse, victims wait until after 20 to 50 violent incidents before contacting police. The risk of lethality to first responders doubles when domestic violence incidents are also marked by animal abuse (Campbell et al. 2018).
- 32% of domestic violence survivors in shelters reported their children had also harmed animals, repeating the intergenerational cycle of violence (Ascione 1998).

### 4.2. Animal Abuse and Child Maltreatment

- Cruelty to animals is one of the earliest symptoms of conduct disorder, showing up at the age of $6\text{-}^1/_2$ (Frick et al. 1993).
- Childhood witnessing of animal cruelty results in significantly more risk of adolescent or adult interpersonal violence (DeGue and DiLillo 2009).
- 43% of school shooters were reported to have histories of animal cruelty (Verlinden et al. 2000).
- Youths who bully others—and those who have been bullied—are at increased risk for committing animal abuse (Baldry 2005; Gullone and Robertson 2008; Henry and Sanders 2007; Vaughn et al. 2011).
- 60% of families under investigation for child abuse, and 88% for physical child abuse, reported animal cruelty. Two-thirds of these cruelty incidents were perpetrated by the adult males; one-third by the children (DeViney et al. 1983).
- 62–76% of animal abuse in the home occurs in the presence of children, causing emotional distress (Faver and Strand 2003).
- Sexually abused children are five times more likely to abuse animals (Ascione et al. 2003).
- Children's committing animal abuse may be an indication of the child abuse they have suffered as well as an indicator of future deviant behavior (Hoffer et al. 2018).

### 4.3. Animal Abuse and Elder Abuse

- More than one-third of APS caseworkers reported that their clients' pets are threatened, injured, killed, or denied care. Furthermore, 75% reported that their clients' concerns for their pets affected their decisions to accept interventions or other services (Boat and Knight 2000).
- In one study, 92% of adult protective services caseworkers reported they encountered animal neglect co-existing with their clients' inability to care for themselves (Raymond 2003). Yet few agencies report having working relationships with community animal care and control agencies (Hoy-Gerlach and Wehman 2017).

### 4.4. Animal Sexual Abuse and Other Crimes

- 11% of individuals convicted of having sex with animals had prior convictions for child pornography. In 5% of these arrests, animal pornography had been used to groom a child for sexual behavior (Edwards 2019).
- Among 1248 sexually violent predators in Virginia, 2.6% had a history of engaging in bestiality, particularly if they had been victims of childhood sexual abuse themselves (Holoyda et al. 2020).
- Among 44,202 men being evaluated for sexual misconduct, 28% had committed a sexual offense against a child, and 5% reported a sexual interest in bestiality. Among the child sex offenders, bestiality was found to be the single largest factor in predicting

increased risk to molest a child, particularly if sexual interest or contact with animals began at an earlier age (Edwards 2019).

Many of these cases can be described as part of a predictive graduation hypothesis whereby people who harm animals need greater and greater thrills and escalate into crimes against humans. However, Arluke et al. (1999) reported that a general deviance theory—in which animal abuse is part of a constellation of behaviors committed by individuals who are more likely to commit other forms of deviance which can precede or follow the animal cruelty incident—is a more prevalent scenario. This has been supported by other studies, most notably by the FBI's Behavioral Analysis Unit (Hoffer et al. 2018).

## 5. Link-Based Criminological Advancements

This new focus on animal abuse as neither an animal rights nor animal welfare issue, but rather one which also impacts human well-being, has resulted in dramatic improvements in legislation and organizational programming aimed at protecting both human and non-human members of society. The Link offers a pragmatic perspective which is more readily embraced by legislators for whom animals' moral concerns or alleged rights have not been priorities. Recent developments in this arena include:

### 5.1. National Data Collection

In 2016, recognizing the absence of any state or national databases quantifying the prevalence of incidents of animal maltreatment, the Federal Bureau of Investigation incorporated four distinct types of crimes against animals within its new National Incident-Based Reporting System (NIBRS). This database, employed by thousands of law enforcement agencies across the U.S., now began to log incidents of Intentional Animal Abuse and Torture, Simple Neglect and Gross Neglect (i.e., animal hoarding), Organized Animal Abuse (i.e., dog- and cock-fighting), and Animal Sexual Abuse. Previously, any offenses against animals had not been specifically identified and were lumped into an "all other offenses" classification in the FBI's Uniform Crime Reporting program, which had been in use and unchanged since 1930. Significantly, these four crimes are housed in NIBRS in Group A as crimes against society, rather than in the more traditional category of crimes against property (DeSousa 2017).

"Some studies say that cruelty to animals is a precursor to larger crime", said Nelson Ferry, who then worked in the FBI's Criminal Statistics Management Unit, which manages NIBRS. "That's one of the items that we're looking at", added John Thompson, then deputy executive director of the National Sheriffs' Association which strongly advocated for the FBI to include animal cruelty data, "If somebody is harming an animal, there is a good chance they also are hurting a human. If we see patterns of animal abuse, the odds are that something else is going on" (FBI Federal Bureau of Investigation).

"With this information, law enforcement and victim services would be able to better target their intervention efforts with respect to both animal cruelty and those crimes for which animal cruelty serves as a marker", said Dr. Mary Lou Randour of the Animal Welfare Institute, which worked closely with the National Sheriffs' Association to advance their cause. "Identifying and analyzing animal cruelty crimes would provide an important tool for law enforcement" (FBI Federal Bureau of Investigation).

### 5.2. Protecting Animals and People in Domestic Violence

There is substantial empirical and anecdotal evidence linking acts of intentional animal abuse specifically intended to intimidate intimate partners as a method of coercive control that serves to prevent many victims from leaving abusive situations. Animal abuse normalizes a culture of violence through the abuser's perverse satisfaction of hurting an animal as a means to instill fear and intolerance for rules being broken and as an orchestrated punishment upon a proxy for what is perceived to be another family member's unsatisfactory behavior (Roguski 2012). The depth of emotional attachments to these pets, particularly among women and children in homes marked by domestic violence and child

abuse, frequently enable batterers to target animals with threatened or actual cruelties as a form of emotional blackmail (Arkow 2014).

For example, a study by the Urban Resource Institute and National Domestic Violence Hotline (2021) reported that 97% of hotline callers said that keeping their pets with them is an important factor in deciding whether to seek shelter; 50% would not consider shelter if they could not take their pets with them; 48% feared the abuser would harm or kill the pets; 30% said their children had witnessed or been aware of abuse or threats to a pet; and 91% indicated that their pets' emotional support and physical protection are significant in their ability to survive and heal.

As many as 71% of domestic violence survivors reported their partners killed, harmed, or threatened animals as a means of demonstrating authority (Ascione et al. 1997). These animal abuse incidents frequently were vivid demonstrations or opaque intimations with an expressed or implied message that whatever was done to the animal would be repeated against the human victim unless she complied. In one study, 87% of the animal cruelties were committed in the presence of women, and 75% in the presence of children, in deliberate plots to instill emotional damage and to hold the entire family hostage or to extract revenge (Quinlisk 1999).

Research has found that batterers who abuse pets are more dangerous than those who do not: pet-abusing batterers employ more controlling behaviors, particularly sexual violence, marital rape, emotional violence, and stalking (Simmons and Lehmann 2007). Other studies have revealed that the four most significant risk factors for becoming a domestic violence abuser are lack of a high school diploma; substance abuse; fair or poor mental health; and a history of threatened or actual pet abuse (Walton-Moss et al. 2005). A study of calls into a municipal domestic violence crisis line determined that the three greatest risk factors for lethality were access to weapons, threats of suicide, and threats to mutilate or kill the family pets (Arkow 2014).

In response, innovative legislation is addressing these linkages. At the time of writing, 10 states have laws defining acts of intimidating animal abuse in an intimate partner violence setting as acts of domestic violence, paving the way for them to be prosecuted as such as well as under animal cruelty statutes. At the time of writing, 37 states empower courts to issue domestic violence protection-from-abuse orders that specifically include pets and, in some cases, livestock. Four states, recognizing the contentious nature of emotional attachments to animals in bitter divorce cases, allow judges to award custody of companion animals in what the court deems to be the animals' best interests, mirroring long-standing provisions affecting child custody.

Similarly, major programmatic advancements have facilitated the ability of human victims and their pets to escape abusers. At the time of writing, some 250 domestic violence shelters in the U.S., and several more in other countries, are pet-friendly with facilities for keeping limited numbers of companion animals, thereby removing one significant barrier for families seeking safety (Sheltering Animals and Families Together 2021).

### 5.3. Cross-Reporting

An earlier recognition of incipient animal, child, and elder abuse should theoretically be advanced by a more systematic structure of statutorily mandated cross-reporting among child, adult, and animal protective agencies. To date, such cross-reporting provisions are sporadic (National Link Coalition 2021d); at the time of writing, only five states (Connecticut, Florida, Illinois, Ohio, and West Virginia) have full two-way cross-reporting where animal control and humane officers are required to report suspected child abuse and child protective workers are mandated to report suspected animal abuse. In ten other states, such reporting is either one-way or permissive rather than mandatory. Cross-reporting between animal care and control agencies and adult protective services is even more limited, with only 10 states requiring or permitting some forms of cross-reporting. However, the effectiveness of such cross-reporting is unknown. Considerable research is needed to document the number and resolution of cases of child, elder, and animal

abuse that are cross-referred by animal care and control, child protective services, and adult protective services, and to identify mechanisms and funding streams which would facilitate the systematization of such processes.

### 5.4. Felonization of Animal Cruelty

The traditional perspective of animal issues being less serious than human welfare concerns resulted in most animal welfare crimes being treated as misdemeanors, petty offenses, or disorderly persons offenses. Beginning in the early 1990s, when only five U.S. states had statutes defining acts of animal abuse as felonies, growing awareness that animal abuse is also a human safety concern helped inspire the expansion of animal cruelty legislation. Today, all 50 states have some form of anti-cruelty legislation—often dogfighting or animal sexual abuse—categorized as felonies and prosecutable accordingly.

While increased penalties and the potential for extensive terms of incarceration have not yet been demonstrated to be a panacea or even deterrents to violence against animals, making egregious crimes felonies gives the criminal justice system additional resources and opportunities to hold offenders accountable. They further make a public statement that these laws are finally being taken seriously and have helped fuel other recent developments, such as the rise of veterinary forensics as a specialty to enhance the success of prosecutions and the increased number of specialized animal cruelty units in prosecutors' offices and task forces in law enforcement agencies.

It should not be inferred that increased recognition of the seriousness of animal cruelty cases by legislators, law enforcement, prosecutors, and judges automatically will result in greater incarceration of offenders; prison sentences for offenders are still exceedingly rare and generally reserved for the most egregious cases, often those which include human victims as well. Nor should it be assumed that animal abuse is the most reliable predictor of human violence. Considerable additional research is needed to examine the most effective means of preventing animal abuse and responding to those who commit these crimes.

### 5.5. Animal Sexual Abuse

Although sex with animals has long been condemned as an abomination under Christian morality and has even been a capital offense, such as occurred in Sweden in the 13th and 14th centuries where both humans and animals could receive a death sentence (Lönngren 2020), in contemporary rural American communities, sex with animals was often trivialized and considered a social norm. Where such laws prohibiting sex with animals exist, they are often defined as "bestiality", "crime against nature", or "unnatural acts". Growing evidence links sex with animals and online animal pornography with high correlated incidence with child sexual abuse, child pornography, and other sex crimes (Edwards 2019). At the time of writing, 48 states have enacted laws prohibiting sex with animals; several of these laws model interpersonal criminology by redefining these incidents as animal sexual abuse or animal sexual assault.

In 2019, the federal government closed a loophole in the 2010 Animal Crush Video Prohibition Act, which had made the creation, sale, and distribution of animal pornographic videos illegal, by enacting the PACT (Preventing Animal Cruelty and Torture) Act, which now makes the underlying acts depicted in the videos also illegal. Under the PACT Act, these acts are now federal felonies with offenders facing fines and up to seven years in prison.

### 5.6. Veterinary Forensics

Growing interest in prosecuting crimes against animals, driven in part by the emergence of animal law curricula in over 160 law schools (Animal Legal Defense Fund 2021a) and prosecution task forces being established to cope with the volume and demanding specialized intricacies of animal cruelty cases, gave rise to a need to develop "Animal CSI"—specialized animal cruelty crime scene investigations modeled after techniques long employed by law enforcement in solving human crimes. A growing corpus of professional

literature and professional continuing education courses are available to train veterinary professionals, prosecutors, and humane investigators in detailed crime scene analysis, evidence collection and preservation, and case management.

## 6. A Mystery Left Unsolved

The enforcement of animal cruelty laws, whether by nonprofit humane societies and SPCAs or municipal animal control or animal services officers, has traditionally had a singular focus of rescuing and removing animals from abusive or deleterious conditions with little regard for whether ancillary criminal offenses might be present or if the alleged animal abuse is a criminological antecedent to incipient violent or antisocial behavior. However, the Link between animal abuse and other crimes has a long historical tradition, often overlooked. American colonies enacted anti-cruelty laws as early as 1641 in Massachusetts; a mere eight years later, courts in that colony prosecuted two separate cases in which an individual was alleged to have beaten both a person and an animal (Arkow 2019).

Similarly, the Link between animal abuse and human violence has deep roots in public thought. Perhaps the most popularized early expression of animal abuse as a potential precursor to interpersonal violence were the series of four copperplate engravings by the 1750′s British artist William Hogarth. His "Four Stages of Cruelty" depict a fictional villain Tom Nero (the name may represent a contraction of Tom, he is no hero) (Smith-Blackmore 2017). As a youth he is seen torturing animals; as a young man he whips a horse with a broken leg; he then moves on to be a callous murderer of his pregnant lover and finally as a convict who has been sentenced to death. In a final act of irony, a dog similar to the one he abused in childhood is seen eating his entrails as he lies on the dissection table. All along his journey in the four panels are a Bruegel-like array of other members of society who are committing other social ills and cruelties. Hogarth's depictions clearly demonstrated the popularized association between animal cruelty and other forms of violence, what we now refer to as "The Link".

Whereas in earlier years the difference between animal welfare and animal control was described (Arkow 1987) as two sides of the same coin, with animal welfare protecting animals from people and animal control protecting people from animals, the Link refines this concept into a broader, more ecological perspective. Today we see humane societies and animal control agencies as the same side of the coin, albeit with different funding strategies, each having a common purpose of protecting both vulnerable humans and vulnerable nonhumans under a broader umbrella of community violence prevention (Arkow 2019). This philosophical change was described in a market study of animal shelters by Ipsos-Reid:

> "The philosophy in the animal welfare community is switching to addressing human problems that underlie crises with animals. Animal shelters' service philosophy is evolving to recognize that treating symptoms of animal welfare problems, such as animal homelessness, abuse and neglect, is only a stopgap solution: to be truly effective, underlying causes such as community and family dysfunction and violence must be addressed" (PetLynx 2011)

The concept of humaneness is subjective, culturally defined, and shifts over time. Scobey and Graham (1970) wrote,

> "We all know, and yet none of us knows, what humaneness is. What is defined by members of a social group as humane may change as individuals seek and acknowledge new data with regard to humanness" (p. 1)

The origins of the generic term "humane society" in conjunction with animal welfare remain shrouded. How an organization founded as a human lifesaving service came to be more widely recognized for animal welfare, sheltering, and community education in an American context—while keeping its original connotation for British audiences—is a subject for ongoing discussion and historical research. The author welcomes input from readers who can shed light on this curious etymological mystery.

**Funding:** This research received no external funding.

**Conflicts of Interest:** The author declares no conflict of interest.

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
