# Peer review of "“Humane Criminology”: An Inclusive Victimology Protecting Animals and People"

_socsci, doi:10.3390/socsci10090335_

Round 1
Reviewer 1 Report
This is a wonderfully written, lucid exposition of an interesting topic. The history section in particular is well done and makes a nice contribution to the literature.
I do worry about the omission of any engagement with the growing and important literature on anti-carceral animal law. It is strange to have a paper fashioned as a criminology project about animal law that does not, for example, engage with the 2019 book, Beyond Cages. This omission is rather striking in light of the fact that said book contains an entire chapter criticizing what the author calls "link-think" or the hasty conclusion that more incarceration or convictions will cure the presumed link between animal and human violence. The Link research (missing as it is longitudinal studies and plagued by some study designs) is imperfect. More should be done to note this fact. It is not necessary to discount the link, but there are many, many links and this is but one and the research here is imperfect.
Closely related, even if one wants to take the link studies at face value, it would be a serious mistake to assume without data that more policing and prosecution will cure or remedy the link. The idea that a deeper connection with criminal prosecutions will reduce animal crime is unproven, and the assumption that it will lead to persons not committing crimes against persons is pretty dramatically undermined by other criminology data.
In short, I think more can be done to note that the link is not infallable, and may not be the most reliable predictor of human violence. That does not mean that animal abuse should be ignored or de-emphasized. But in 2021 the reductive idea that there is one link, and the way to cure it is more prosecutions should be challenged. I think the author(s) can be more nuanced on this point.
The author clearly has a perspective, which is fine and valuable. But it should not be so stridently presented as to ignore contrary literature. The paper agues that legislation like more felony laws are "pragmatic" advances in law and policy and that they are obtainable because such reforms are "more readily embraced by legislators." But this ignores a feminist literature (like the book The Feminist War on Crime) that painstakingly show that these felony type laws are not always good for the victims -- it is often supported by legislatures precisely because it does not undermine the status quo. Punishing animal crimes that are defined as felonies (and exempting farm abuse) may not be doing anything to change social values about animals, but rather simply reinforcing the old trope that we Americans can solve any problem through more crime and punishment. I would strongly urge a more nuanced approach that recognizes and acknowledges the criminology research showing that the criminal prosecutions celebrated in this paper are, in the view of some, criminogenic. The author need not agree, but the failure to engage with these literatures at this point in history is too reductive.
Author Response
Thanks for your constructive comments – they are appreciated!
I was not familiar with the book “Beyond Cages” mentioned by you; I do know, however, that that book’s author is somewhat controversial in the animal care and control and animal welfare prosecution fields and that his arguments are not yet widely accepted. It would be inappropriate for me to include any reference to this work as I am not terribly familiar with its specific propositions. Incarceration of animal cruelty offenders is extremely rare, generally reserved for only the most egregious of cases (including those involving child sexual abuse and human violence) and is not relevant to the still-unproven argument that animals have inherent legal rights. There is instead a movement within the animal care and control field to try to reduce the “incarceration” of companion animals by teaching pet owners positive-reinforcement animal behavior modification techniques to reduce the number of animals which would otherwise be surrendered to shelters for solvable behavior problems.
Meanwhile, as much as it could be argued that rehabilitation is better than incarceration, the reality is that there are few viable alternatives to criminal penalties. One would be behavioral interventions; to date, none of the few such psychological programs that have been developed have been evaluated to demonstrate what effectiveness, if any, they have in preventing recidivism. Another alternative would be empathy-building community service to have animal cruelty offenders work off their sentences in animal shelters; this tactic has been widely condemned by the animal rights community who fear what these offenders might do if exposed to animals, similar to child sex offenders being barred from working with children.
Likewise, the “Beyond Cages” author recommends a greater focus on “animal rights” as an emblem of social oppression. The focus of this article is neither on social oppression nor on entrenching a divide between human and non-human animals. It’s possible other articles for this special issue will address those topics. Rather, I would argue that showing the common origins of child and animal protection and concern for humane values bridges the human/non-human animal divide.
In addition, because many legislators see animal rights activists as disruptive and because animals have no legal rights, nor are they likely to in the foreseeable future, a much more pragmatic strategy is to demonstrate The Link between animal abuse and human violence. This message avoids the animal welfare/animal rights debates which tend to disrupt legislative hearings and appeals to legislators for whom animals are not a moral concern and to reluctant law enforcement agencies as public safety is a matter of their concern. The Link is not a talisman or shibboleth as “Beyond Cages” would suggest: it is a practical strategy to be incorporated within other domains including social work, behavioral health, animal behavior, veterinary and human medicine, domestic violence, child maltreatment, and elder abuse.
Similarly, neither are incarceration nor convictions an ultimate panacea and solution to violence against animals. While it has not been demonstrated empirically that felonization and greater fines for animal cruelty serve as a deterrent, intensifying criminal penalties for animal abuse, backed up by the growth of veterinary forensic techniques and an increasing number of animal cruelty task forces in law enforcement agencies and units in prosecutors’ offices, make a public statement that these laws are finally being taken seriously and give the criminal justice system additional resources and opportunities to hold offenders accountable. While research demonstrating these links is perhaps not as empirically robust as in other fields and admittedly there is little if any research (yet) demonstrating the efficacy of longer incarceration periods, I have added information in Sec. 5.4 describing the need for more research into the effectiveness of incarcerating animal cruelty offenders and that the link is not infallable, and may not be the most reliable predictor of human violence. (I state repeatedly that animal abuse is only a “potential” indicator or predictor of interpersonal violence.)
The exemption of agricultural animal abuse is indeed a horrendous omission, but until the animal welfare community musters the same financial and political clout as the agribusiness lobby, I fear change will not come to this aspect of animal cruelty enforcement and prosecution. The entire history of the humane movement can be seen as an urban mindset trying to impose its values of animals as autonomous beings on rural communities who see animals as commodities rather than as companions – replacing attachments to individual companion animals with the economic imperatives of herd management, if you will. I’ve addressed this in the revised opening paragraph.
Regarding your Feminist War or Crime concern that incarceration only leads to further re-victimization: one of the fundamental differences between the child protection movement and the animal protection movement – despite their common heritage – is that a goal of the former is to reunite the family unit, while the latter aims to remove the victim from the family, which could be perceived as re-victimization. This is a fascinating perspective but beyond the purview of this article.
Reviewer 2 Report
An excellent piece; I've strongly recommended publication.
The abstract is a bit misleading. Overall the article reads as two pieces: one on the etymology of humane and one (much more valuable) on cruelty, crime & regulation
Lines 23 through 28. Criminology also covers the criminal activity of corporations. It is not restricted to individuals. Note also that although non-human animals were on occasion considered to have engaged in crime and were accordingly punished for that activity, non-human animals in contemporary liberal democratic states are deemed to have no criminal responsibility and more broadly not to be legal persons. Piers Beirne's work (esp debunking some myths about nonhuman animal criminals) is valuable. Keith Thomas's luminous Man & the Natural World might support yr reference for St Thomas,. Porphyry, Montaigne, St Francis etc
Lines 91 through 92. Not sure about the “humane movement” and its restriction to North America. Use of “humane” in relation to non-human animals is now common in Australia and New Zealand, in part through import of language from the US by animal rights activists over the past thirty years
Line 95, 532, 641. Typo
Line 96. Jeremy Bentham on cruelty to non-human animals as a wrong per se, irrespective of the impact on humans, was influential
Line 792. Reference to trivialisation is misleading and disregards the history of capital punishment for bestiality (often conflated with sodomy) in Europe, N America and Australasia over several centuries. Jens Rydstrom on Scandinavia might be useful. Other sources claim 600 executions in Sweden alone from 1630 to 1770 and there were hangings, burnings etc in Scotland, Germany, England. Several hangings in early colonial Australia re intercourse with pigs and dogs
Author Response
Thanks for your comments and constructive ideas – they are appreciated!
Lines 23 through 28. You are correct that criminology also covers the criminal activity of corporations. I have edited the text accordingly. The exemption of agricultural animal abuse in particular is a horrendous omission, but until the animal welfare community musters the same financial and political clout as the agribusiness lobby, I fear not much will change, hence the focus on individual acts of animal cruelty, abuse and neglect.
Lines 91 through 92. The use of “humane” in relation to non-human animals may now be more common in Australia and New Zealand, in part through import of language from the US, but I’ve yet to find a significant number of Australasian animal shelters calling themselves humane societies. I will note this importation in lines 225-227.
Line 95, 532, 641. Typos. Good catches! I’ll correct.
Line 96. Jeremy Bentham on cruelty to non-human animals as a wrong per se, irrespective of the impact on humans, was influential. I believe I addressed this in lines 488-493.
Line 792: Subsequent to the submission of this MS, I came across some of the Scandinavian literature you noted. I’ll add this in, noting that in some cases it was a capital offense – and in others accepted as a cultural norm..